# Refined Estimation of Potential GDP Exposure in Low-Elevation Coastal Zones (LECZ) of China Based on Multi-Source Data and Random Forest

**Feixiang Li** [1], **Liwei Mao** [2], **Qian Chen** [1] and **Xuchao Yang** [1,*]

1   Ocean College, Zhejiang University, Zhoushan 316021, China
2   Hangzhou City Planning and Design Academy, Hangzhou 310012, China
*   Correspondence: yangxuchao@zju.edu.cn; Tel.: +86-0580-2092277

**Abstract:** With climate change and rising sea levels, the residents and assets in low-elevation coastal zones (LECZ) are at increasing risk. The application of high-resolution gridded population datasets in recent years has highlighted the threats faced by people living in LECZ. However, the potential exposure of gross domestic product (GDP) within LECZ remains unknown, due to the absence of refined GDP datasets and corresponding analyzes for coastal regions. The climate-related risks faced by LECZ may still be underestimated. In this study, we estimated the potential exposure of GDP in the LECZ across China by overlying DEM with new gridded GDP datasets generated by random forest models. The results show that 24.02% and 22.7% of China's total GDP were located in the LECZ in 2010 and 2019, respectively, while the area of the LECZ only accounted for 1.91% of China's territory. Significant variability appears in the spatial-temporal pattern and the volume of GDP across sectors, which impedes disaster prevention and mitigation efforts within administrative regions. Interannual comparisons reveal a rapid increase in GDP within the LECZ, but a decline in its share of the country. Policy reasons may have driven the slow shift of China's economy to regions far from the LECZ.

**Keywords:** low elevation coastal zones; GDP; climate change; multi-source data; random forest; China

## 1. Introduction

Low-elevation coastal zones (LECZ) are areas below 10 m elevation with hydrological connections to the sea [1], and are widely distributed in coastal zones around the world [2]. Given the advantages in transportation, resource and terrain, such coastal zones are often attractive for residential, commercial and tourism development [3]. However, climate change has accelerated sea level rise (SLR), and intensified tropical and extratropical cyclones, exposing the LECZ to an increasing risk of natural disasters [4–6]. In the extremes, SLR in the 21st century could even exceed 2 m, threatening to inundate large numbers of people and significant amounts of property worldwide [7,8]. In response to this situation, the research on climate change and the population living in LECZ has been of increasing interest in recent years [9–11]. McGranahan et al. [1] first reported that LECZ cover 2% of the world's land area but comprise 10% of the global population. Fine-grained global population datasets, such as LandScan and WorldPop, currently the two most widely known population datasets worldwide, have also been used to estimate the potential exposure of the population in the LECZ across the world [12,13]. However, despite being one of the most important hazard-affected elements in disaster risk assessment, the geographical distribution of the gross domestic product (GDP) in the LECZ has been poorly studied, resulting in an insufficient understanding of the impact of climate change on the economy within LECZ.

Gross domestic product is the primary indicator for measuring economic activity [14,15], and highly urbanized areas are typically densely populated and economically developed [16]. With 3351 cities and 13 megacities globally located in coastal areas [17], substantial GDP will be at risk of inundation from sea level rise by 2100 [18]. In addition, differentiated economic activity characteristics divide the GDP categories, with the primary sector ($GDP_1$) covering agriculture, forestry, livestock and fisheries, the secondary sector ($GDP_2$) including mining, manufacturing and transportation and the tertiary sector ($GDP_3$) containing all service industries. This further highlights the threats faced by GDP within the LECZ, making it difficult for GDP to formulate a unified disaster prevention policy such as population [19]. For example, when the basic industry in the secondary sector suffers from a natural disaster, industrial production, the surrounding natural environment and the population will be continuously affected [20]. Previous studies have also shown that there is an inverted U-shaped relationship between natural disaster losses and wealth, with adequate prevention support leading to losses increasing and then decreasing as wealth increases [19,21]. The geographical analysis of the GDP distribution within the LECZ will be a key step in both disaster prevention and loss assessment for complex economic activities.

Gross domestic product data are usually collected from the administrative level (cities, counties, etc.), which leads to issues such as low resolution and insufficient spatial heterogeneity and hinders their integration with other geospatial data [22,23]. Decomposing traditional GDP statistics with other auxiliary data is a common method for obtaining gridded GDP data [24,25]. Nighttime lights (NTL) are viable indicators for detecting economic activities and are widely used in GDP mapping [26,27]. However, the GDPs of secondary and tertiary sectors in complex urban areas are difficult to differentiate by considering only NTL data [28]. Moreover, for agricultural activities with marginal or no NTL emissions, NTL data are not able to disaggregate GDP from the administrative district scale [29]. The inclusion of multisource data in GDP mapping has become a consensus among researchers [30,31]. In recent years, advances in geospatial big data and machine learning have provided new opportunities for fine-grained GDP mapping [32,33]. Rich economic activity information is embedded in point of interest (POI) and road network data in the era of geospatial big data [28,34]. Machine learning methods are used to effectively capture the nonlinear relationship between covariates and GDP, generating a high-precision GDP distribution map of urban areas [35].

Combining these developments, Chen et al. [28] mapped the distribution of different sectors in mainland China in 2010 at the 1 km resolution. Wang et al. [36] also used machine learning methods to generate global GDP distribution at the 1 km resolution. Nevertheless, it is still difficult to apply previous GDP datasets to the current risk assessment of LECZ due to their outdated data vintage. Also, single-year data hardly allow researchers to see the changes in the potential exposure of GDP, and the impact of government decisions, and make reasonable recommendations. Furthermore, gridded GDP datasets at the 1 km resolution are still too coarse for estimating GDP exposure within the LECZ and could lead to overestimations compared to 100 m resolution [37,38]. Gross domestic product density maps should be updated on both temporal and spatial scales to meet the spatial analysis assessment requirements of GDP in LECZ.

In this study, we took the LECZ of mainland China as the study area and decomposed the 2010 and 2019 official GDP by integrating multisource remote sensing data and geospatial big data using the random forest (RF) algorithm. The years 2010 and 2019 exclude the economic impact of COVID-19, etc., and have complete official statistics in the study area. The $GDP_1$, $GDP_2$ the $GDP_3$ were modeled, respectively, and gridded GDP maps of different sectors with a resolution of 100 m were produced. Then, a spatial analysis of the potential exposure of GDP in LECZ across China was performed based on digital elevation model (DEM) data and our GDP gridded maps. Finally, we contrasted the risks posed by GDP and population in LECZ and emphasized the importance of assessing GDP within the LECZ and the trends of potential exposure of GDP and population within LECZ in China.

## 2. Materials and Methodology

### 2.1. Data Source and Preprocessing

This study uses the logarithm of county-level GDP density as the dependent variable for 11 coastal provinces in China, from north to south, namely Liaoning, Hebei, Tianjin, Shandong, Jiangsu, Shanghai, Zhejiang, Fujian, Guangdong, Guangxi and Hainan (Figure 1). Hong Kong, Macau and Taiwan were excluded because of their distinct political and economic statuses. Official GDP data were obtained from the China Statistical Yearbook, with a total of 822 counties in 2010 and 806 counties in 2019. GDP1, GDP2 and GDP3 were defined according to official Chinese National Economic Classification of Industries (No. GB/T 4754-2017), and their statistics were collected at county level and spatially joined to the corresponding administrative boundaries, respectively. According to the official statistics, the abovementioned regions accounted for about 58.36% of China's GDP in 2019.

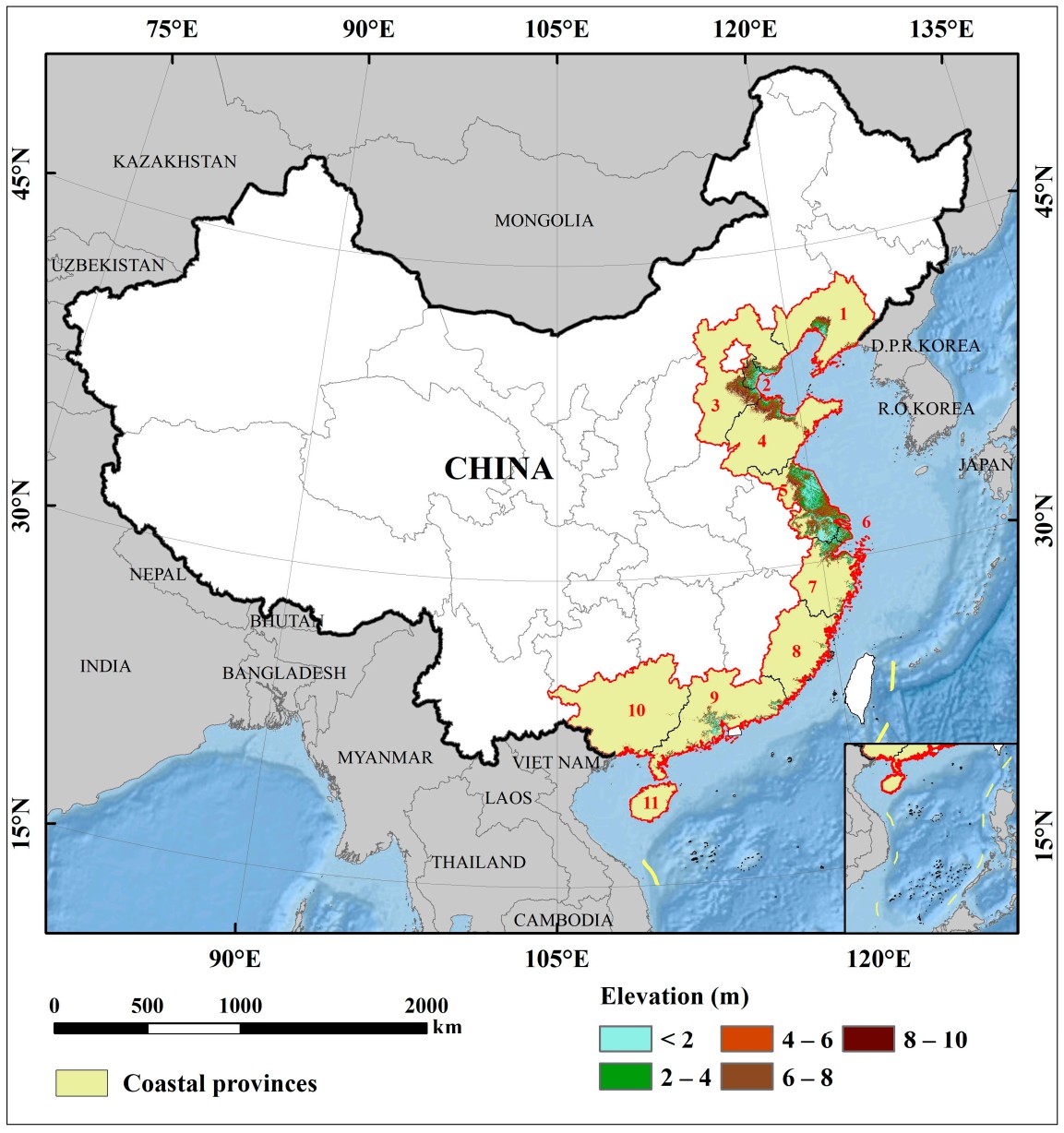

**Figure 1.** Coastal provinces and LECZ in China. Numbers 1–11 represent Liaoning, Tianjin, Shandong, Jiangsu, Shanghai, Zhejiang, Fujian, Guangdong, Guangxi and Hainan, respectively.

The geospatial big data used in this study include POI data and road network data, all of which were extracted from Baidu Map (https://map.baidu.com (accessed on 23 August 2021)), one of the largest desktop and mobile map service providers in China [39,40]. Road networks and POIs have been widely used for mapping of secondary and tertiary industries in previous studies [33]. According to the Chinese semantic description of POIs, we divided POIs into secondary sector-related POIs (factories, mining companies, etc.) and tertiary sector-related POIs (commercial buildings, restaurants, banks, etc.) [28]. Then, kernel density estimation [41] was used for the conversion of discrete POI data into continuous and smooth density surfaces (POIs-Sec-density and POI-Ter-density) with a bandwidth of 200 m, an optimal size for urban building function identification and recognition [42]. The road network data were used to generate the corresponding distance to the nearest road (DtN-road) layers at a resolution of 100 m.

The NTL data for 2010 and 2019 were obtained from the extended time-series of the global NPP-VIIRS-like NTL dataset [43]. This NTL dataset was calibrated from the DMSP-OLS and NPP-VIIRS NTL data with a time range of 2000–2020 and a spatial resolution of 500 m.

Land surface temperature (LST) data were derived from the MODIS MOD11A1 product (https://ladsweb.modaps.eosdis.nasa.gov/search/ (accessed on 13 January 2022)), which records global daily mean LST information and separately stores daytime and nighttime data at the 1 km resolution. The mean daytime LST (LST-day) and mean nighttime LST (LST-night) for the summers of 2010 and 2019 were calculated using LST images [44].

The net primary product (NPP) data were obtained from the MODIS MOD17A3H product in 500 m resolution. The normalized difference vegetation index (NDVI) data were calculated based on Landsat 8 satellite data at the 30 m resolution. The near-infrared band (band 5) and red band (band 4) of Landsat8 were selected using the Google Earth Engine to calculate the annual average NDVIs in 2010 and 2019.

The land cover datasets for China are the annual data from 1990 to 2010 with a resolution of 30 m produced by Yang and Huang [45]. The proportions of cropland, forest, grassland and water bodies were calculated in 100 m × 100 m grids (e.g., cropland rate, forest rate, grassland rate and water rate).

The 30 m resolution DEM dataset was obtained from the Jet Propulsion Laboratory of the National Aeronautics and Space Administration (https://earthdata.nasa.gov/ (accessed on 12 December 2020)). These DEM data (hereafter referred to as NASADEM) are a reprocessing of the Shuttle Radar Topography Mission data and combine data from Advanced Spaceborne Thermal Emission and Reflection Radiometer, Global DEM and Geoscience Laser Altimeter with excellent accuracy and quality. Finally, the slope layer with 100 m resolution was generated by the NASADEM.

All remote sensing data used in this study are publicly available. All raster data were re-projected to the Albers Conical Equal Area Projection and resampled to 100 m resolution in ArcGIS 10.2 (Hangzhou, China). The Snap Raster function was kept on to ensure that the different covariates were aligned with each other.

### 2.2. Methodology

The workflow of data processing, RF model fitting, dasymetric GDP mapping and LECZ extraction is shown in Figure 2. Ultimately, the assessment of potential GDP exposure within the LECZ is achieved by overlaying the coastal GDP density maps with the LECZ range.

### 2.2.1. RF Model Fitting

The RF algorithm uses the decision tree to create an ensemble of machine learning algorithms [46]. When dealing with complex datasets, RF usually performs better than linear regression models since it can handle nonlinear relationships between variables [47]. Previous studies have also demonstrated the good performance of RF in mapping socioeconomic factors [48,49] and it has been widely used in GDP mapping [33]. In this study, six

RF models for different sectors of GDP in 2010 and 2019 were developed to map reliable coastal GDP distributions.

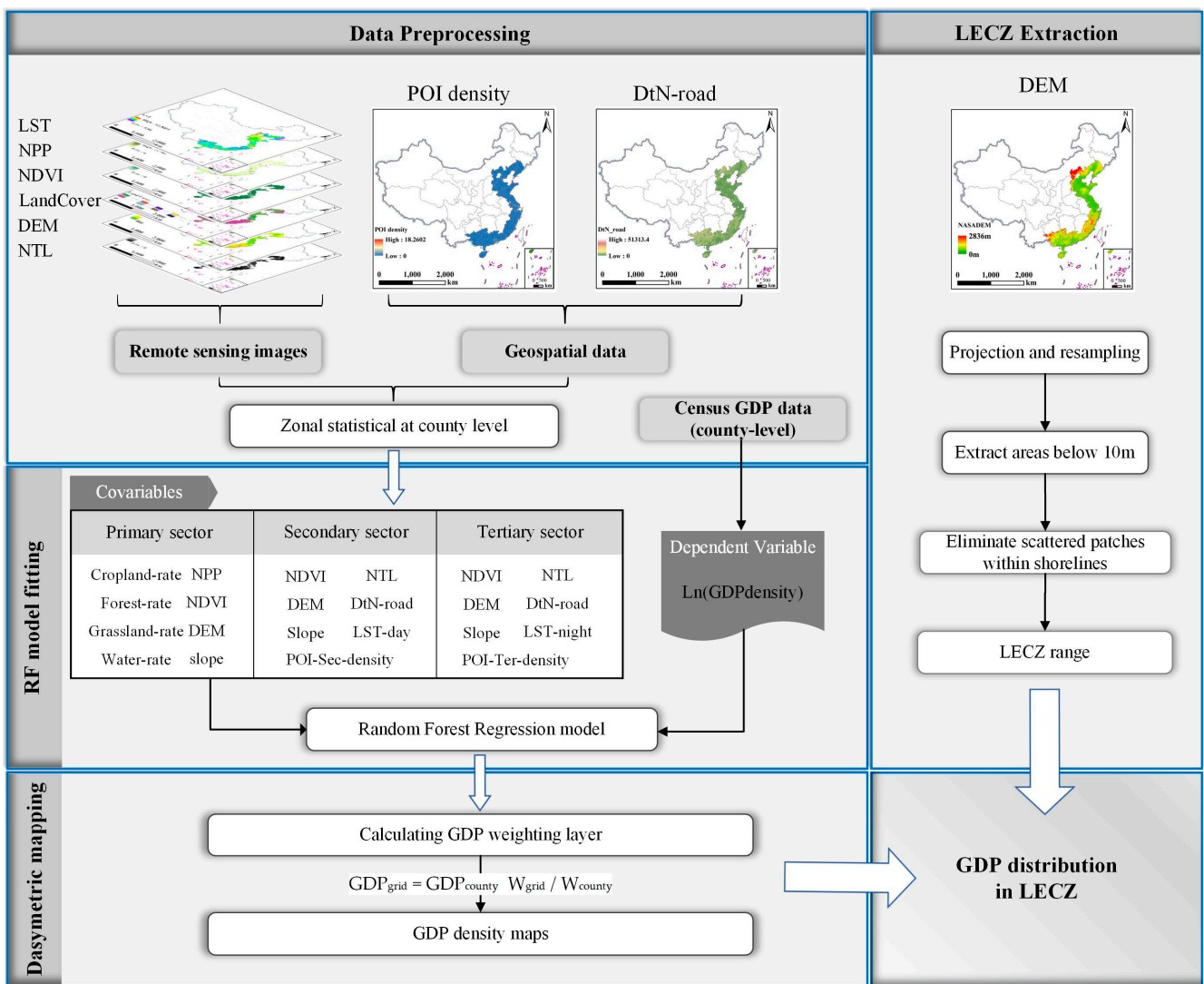

**Figure 2.** Flow diagram for producing the GDP maps of LECZ in China.

For the model estimating $GDP_1$, eight covariables of NDVI, NPP, DEM, slope, cropland-rate, forest-rate, grassland-rate and water-rate were selected, which are potentially related to agricultural productivity. In consideration of the characteristics of the primary sector, built-up areas were given a $GDP_1$ value of 0. For the $GDP_2$ model, seven covariables were selected: NTL, POI-Sec-density, DtN-road, LST-day, NDVI, DEM and slope. For the model mapping $GDP_3$, the covariables were POIs-Ter-density, road-density, NTL, LST-day, DEM, slope and NDVI. Ideally establishing a nonlinear fit of cell-by-cell GDP data to the covariates is difficult due to the lack of field-grid GDP data [28]. Therefore, a prevalent method of data downscaling is to fit this nonlinear relationship using the average of the independent variables and covariates within the administrative region as the training set [48]. In this study, all covariables were aggregated by county and linked with the natural logarithm of census GDP density of the different sectors for fitting with the RF model. After all models were fitted, the covariable data in 100 m resolution were input into the corresponding RF model to produce a GDP-distribution weight layer.

### 2.2.2. Dasymetric Mapping

Dasymetric mapping was used as the approach for revealing a reliable spatial distribution pattern of GDP [50,51]. The official GDP data of individual sectors were disaggregated by weight layers into grids at the 100 m spatial resolution, and the GDP distribution map for each sector for China's coastal provinces was obtained by Formula 1, and finally the total GDP distribution was obtained by summing all sectors (Figure 3).

$$GDP_{grid} = GDP_{county} \times W_{grid}/W_{county} \tag{1}$$

where $W_{grid}$ is the GDP-distribution weight for a 100 m $\times$ 100 m gridded area, $W_{county}$ is the summed GDP-distribution weight of a county, $GDP_{county}$ represents the county census GDP and $GDP_{grid}$ is the distribution GDP for the gridded area.

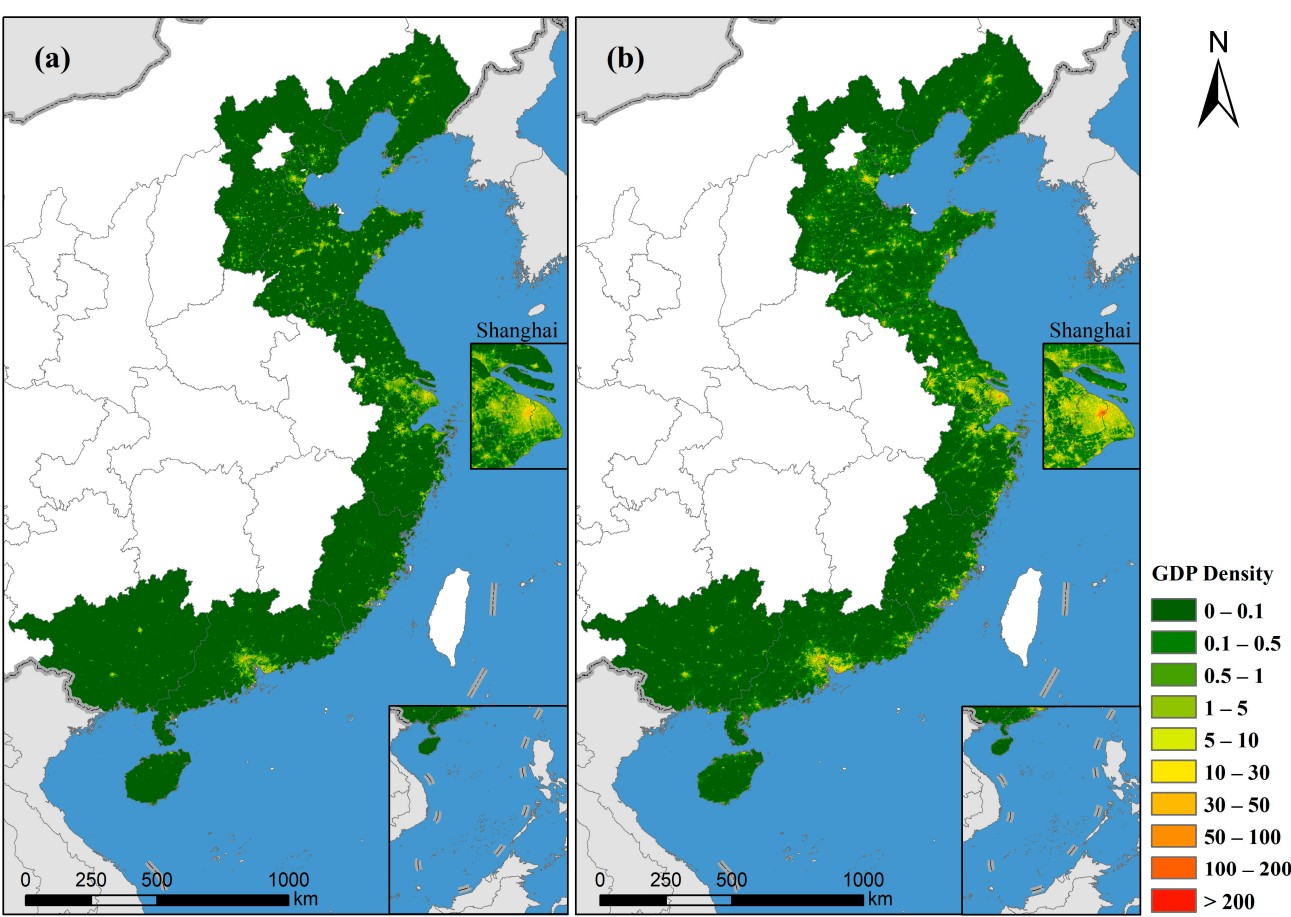

**Figure 3.** Total-GDP distribution of China's coastal areas in (**a**) 2010 and (**b**) 2019 (unit: million CNY/ha).

Due to the general lack of fine-grained real GDP data, the GDP datasets at the pixel scale are difficult to verify. Therefore, referring to previous works [25,28], this study uses the tenfold cross-validation method for county-level samples to evaluate the accuracy of each RF model. All samples were randomly divided into ten pieces, nine of which were selected for modeling and the rest for validation. The evaluation indexes include mean absolute error (MAE), root mean square error (RMSE) and determination coefficient ($R^2$).

### 2.2.3. LECZ Extraction

Low-elevation coastal zones are defined as coastal continuous zones with an elevation of less than 10 m [1]. In this study, the LECZ of mainland China were extracted using NASADEM in ArcGIS 10.2 (Figure 1), and the step can be summarized as follows: (1) Extract

all areas with an elevation of ≤10 m in the study area. (2) Remove scattered patches within the coastline to maintain the continuity between the LECZ and the ocean. Based on the above GDP density maps and LECZ range, the GDP potential exposure maps of LECZ in mainland China with 100 m resolution were obtained.

## 3. Results

### 3.1. Accuracy Assessment of New GDP Datasets

The results of the ten-fold cross-validation of each RF model are shown in Table 1. The RF models of $GDP_3$ have the best performance ($R^2$ = 0.93 and 0.95 for 2010 and 2019, respectively), followed by those of $GDP_2$ ($R^2$ = 0.85 for both),\ and $GDP_1$ ($R^2$ = 0.54 and 0.44). Similar results were obtained in this study compared to Chen et al.'s [28] study for mainland China in 2010. However, it is worth noting that the performance of the $GDP_1$ model appears different on the coast and inland, and the fitting performance is a bit worse, which to some extent explains the necessity of separate modeling for coastal and inland samples. The cross-validation shows that the predictions of the RF model for the three sectors are in good agreement with the logarithm of census GDP density, with MAE in the range of 0.31–0.52 and RMSE in the range of 0.42–0.70. Considering the low share of $GDP_1$ in the coastal economy and the limitation of the land use type, this result can be well applied to the subsequent assessment of the potential exposure of GDP within the LECZ.

**Table 1.** Cross-Validation of the RF Model for Different Sectors.

|         | Year | $R^2$ | MAE | RMSE |
|---------|------|-------|-----|------|
| $GDP_1$ | 2010 | 0.54 | 0.39 | 0.57 |
|         | 2019 | 0.44 | 0.45 | 0.66 |
| $GDP_2$ | 2010 | 0.85 | 0.46 | 0.60 |
|         | 2019 | 0.85 | 0.52 | 0.70 |
| $GDP_3$ | 2010 | 0.93 | 0.33 | 0.43 |
|         | 2019 | 0.95 | 0.31 | 0.42 |

### 3.2. Total-GDP Exposure in the LECZ

The total-GDP distribution of LECZ in mainland China, by overlaying the LECZ range and coastal GDP density map, are presented in Figure 4. The result shows that the total area of LECZ in China is 183,781 km², which accounts for 1.91% of the country's land area. Accordingly, 25.2% of the country's GDP occurred in this region in 2010, amounting to CNY 10,363.8 billion (~USD 1572 billion in 2010). In 2019, the GDP of the LECZ grew by a significant amount, reaching a total of CNY 22,398.8 billion (~USD 3217 billion), whereas its share of the country dropped to 22.7%. The Yangtze River Delta, Pearl River Delta and Bohai Rim are the regions with the most extensive distribution area of the LECZ, as well as the most developed regions on the eastern shore of China. Intensive economic activity has been concentrated in the three economic areas, especially in the major cities of Shanghai, Guangzhou, Shenzhen and Tianjin. The Yangtze River Delta (more specifically Shanghai) has the highest GDP density in its LECZ, with CNY 100 million/ha ( USD 15.2 million/ha in 2010) and CNY 300 million/ha (USD 44.8 million/ha in 2019) in 2010 and 2019, respectively. The fact that more GDP is produced within LECZ significantly increases the exposure to hazards, and the magnitude of their impact on the economy when they occur.

Among all provinces, Jiangsu has the most widely distributed LECZ together with the most GDP in the region, followed by Guangdong and Zhejiang (Table 2). There are also extensive LECZ located in Shandong and Hebei, but compared to the provinces in the Yangtze River Delta, and the Pearl River Delta, the production of GDP in these regions is relatively small. Shanghai has 99.04% of its territory and 88.90% of its GDP in the LECZ, and Tianjin has 88.94% of its territory and 93.1% of its GDP in the LECZ. Although Hainan has only $2.08 \times 10^3$ km² of LECZ, or 6.18% of the territory, 23.1% of GDP is generated within the LECZ.

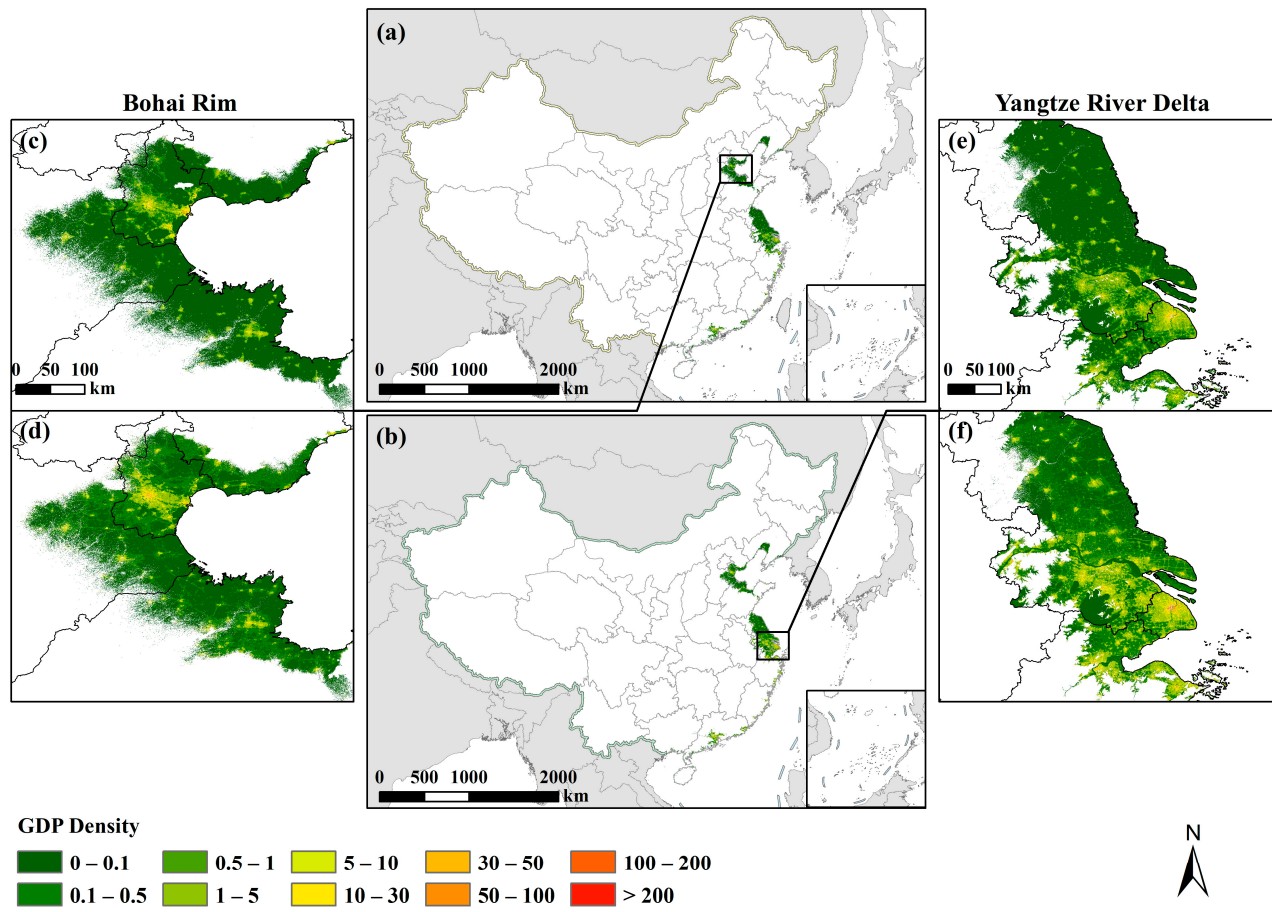

**Figure 4.** Total-GDP distribution of China's LECZ in 2010 (**a**,**c**,**e**) and 2019 (**b**,**d**,**f**) (unit: million CNY/ha).

**Table 2.** GDP statistics in the LECZ of provinces of China.

| Administrative District [a] | LECZ (×10³ km²) | Percentage of LECZ [b] (%) | 2010 | | 2019 | |
|---|---|---|---|---|---|---|
| | | | GDP of LECZ (Billion CNY) | Percentage of GDP (%) [c] | GDP of LECZ (Billion CNY) | Percentage of GDP (%) [c] |
| Liaoning | 11.91 | 8.21 | 298.4 | 16.2 | 377.7 | 15.2 |
| Hebei | 19.43 | 10.39 | 292.8 | 14.4 | 534.7 | 15.2 |
| Tianjin | 10.37 | 89.40 | 953.2 | 93.8 | 1313.6 | 93.1 |
| Shandong | 19.98 | 13.02 | 549.0 | 14.0 | 1025.6 | 14.4 |
| Jiangsu | 69.80 | 69.31 | 2762.9 | 66.7 | 6573.7 | 66.0 |
| Shanghai | 6.59 | 99.04 | 1580.3 | 92.1 | 3392.6 | 88.9 |
| Zhejiang | 18.26 | 17.97 | 1633.9 | 58.9 | 3739.3 | 60.0 |
| Fujian | 3.69 | 3.06 | 388.4 | 26.4 | 1117.4 | 26.4 |
| Guangdong | 20.09 | 11.41 | 1837.0 | 39.9 | 4125.8 | 38.3 |
| Guangxi | 1.57 | 0.66 | 25.8 | 2.7 | 67.0 | 3.2 |
| Hainan | 2.08 | 6.18 | 42.2 | 20.4 | 122.5 | 23.1 |
| China | 183.78 | 1.91 | 10,363.8 | 25.2 | 22,389.8 | 22.7 |

[a] Hong Kong, Macau and Taiwan were excluded because of their distinct political and economic statuses. [b] The proportion of LECZ area to the territory of corresponding administrative district. [c] The proportion of GDP in LECZ to the total GDP of corresponding provinces.

### 3.3. GDP Exposure by Sector in LECZ

The potential exposure of each sector GDP within the LECZ was further assessed and the results are shown in Table 3. Figure 5 shows the distribution of GDP₁ in the LECZ. As opposed to the geographical distribution of total GDP, the areas with relatively high

GDP$_1$ density were mainly located in rural and suburban areas with low urbanization levels. Benefiting from the favorable water conservancy and fishing conditions, the areas surrounding the water bodies tend to have higher GDP$_1$ densities. Overall, the GDP$_1$ is evenly distributed in the urban periphery, with a lower economic output per hectare compared to other sectors. Statistical results show that CNY 429.14 billion (~USD 65 billion in 2010) of GDP$_1$ was generated in LECZ in 2010, accounting for 4.14% of the GDP in LECZ and 11.2% of the total GDP$_1$ of the country. In 2019, the density of GDP$_1$ increased substantially, and a wide range of areas with high GDP$_1$ density emerged in the Yangtze River Delta, with GDP$_1$ in the LECZ reaching CNY 665.47 billion (~USD 96 billion in 2019). However, it is worth noting that the proportion of GDP$_1$ in both the total GDP of LECZ (2.97%) and the primary sector of the country (9.44%) has declined.

**Table 3.** GDP Exposure of Different Sectors and the Proportion of LECZ and Corresponding Sector.

| Exposure Element | 2010 | | | 2019 | | |
|---|---|---|---|---|---|---|
| | Exposed GDP (Billion CNY)/Population | Percentage of LECZ (%) | Percentage of National Total (%) | Exposed GDP (Billion CNY)/Population | Percentage of LECZ (%) | Percentage of National Total (%) |
| GDP$_1$ | 421.1 | 4.1 | 11. 2 | 664.0 | 3.0 | 9.4 |
| GDP$_2$ | 5569.0 | 53.7 | 29.1 | 9408.6 | 42.0 | 24.7 |
| GDP$_3$ | 4373.7 | 42.2 | 24.0 | 12,317.2 | 55.0 | 23.0 |
| total-GDP | 10,363.8 | 100 | 24.02 | 22,389.8 | 100 | 22.7 |
| Landscan | 180,614,045 | / | 13.6 | 211,289,372 | / | 14.9 |
| Yang et al. [38] | 158,208,732 | / | 12.7 | / | / | / |

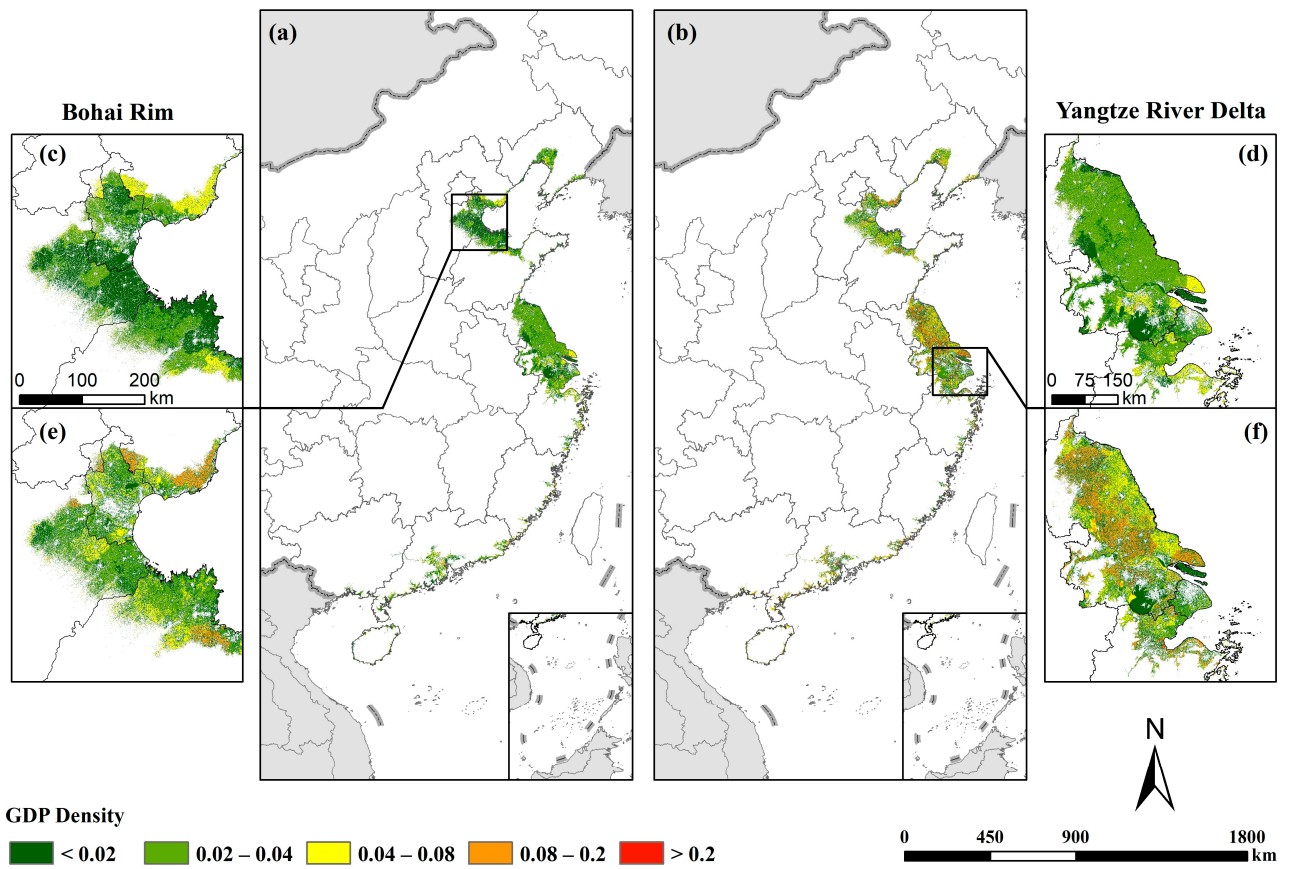

**GDP Density**
■ < 0.02    ■ 0.02 – 0.04    ■ 0.04 – 0.08    ■ 0.08 – 0.2    ■ > 0.2

**Figure 5.** A 100 m gridded GDP$_1$ map of China's LECZ in 2010 (**a**,**c**,**d**) and 2019 (**b**,**e**,**f**) (unit: million CNY/ha).

The spatial distribution for GDP$_2$ in LECZ is more concentrated (Figure 6) and has a higher output value compared with GDP$_1$. In particular, the areas with higher GDP$_2$ density are mainly located in the cities and their suburbs. The Yangtze River Delta and

Pearl River Delta, as the two major urban agglomerations in eastern China, are home to large numbers of secondary industrial enterprises, and the highest $GDP_2$ density occurs in this region. In the Bohai Rims, the high values of $GDP_2$ are mainly concentrated in Tianjin, while the rest of the region shows a sporadic distribution. According to the statistical results of GDP grid data, about CNY 5569.04 billion (~USD 845 billion in 2010) $GDP_2$ was observed for the LECZ in 2010, accounting for 53.69% of the GDP in LECZ and 29.1% of the $GDP_2$ of China. Similar to the growth of $GDP_1$, LECZ experienced a rapid increase in $GDP_2$ from 2010 to 2019, with the output value of $GDP_2$ in urban areas becoming higher and expanding to the periphery. Overall, $GDP_2$ within LECZ increased to CNY 9408.59 billion (~USD 1351 billion) in 2019, despite a significant decline in the percentage of GDP in LECZ (42.02%) and national $GDP_2$ (24.72%).

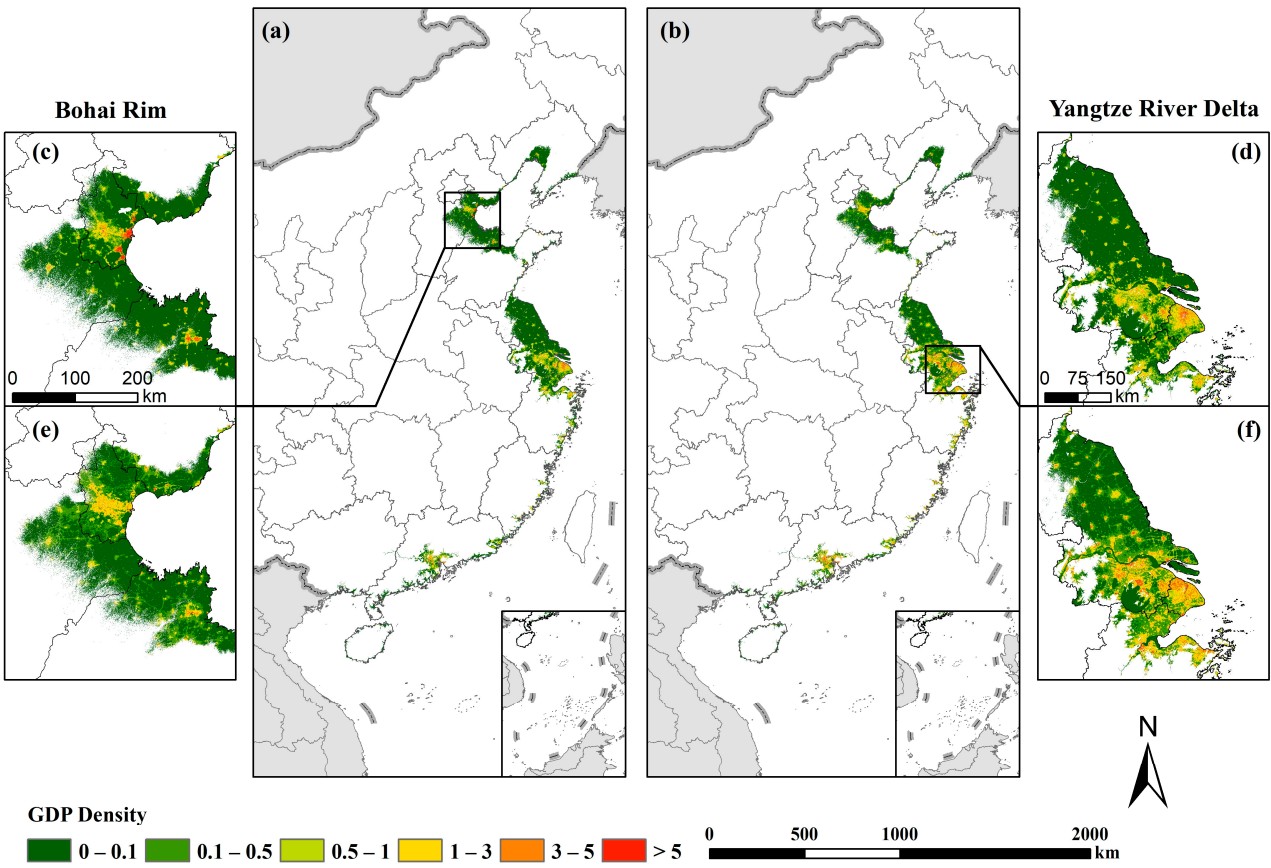

**Figure 6.** A 100 m gridded $GDP_2$ map of China's LECZ in 2010 (**a,c,d**) and 2019 (**b,e,f**) (unit: million CNY/ha).

Geographically, $GDP_3$ exhibits a higher degree of agglomeration than $GDP_2$, concentrated in urban centers (Figure 7). The high-density areas of $GDP_3$ within the LECZ expanded rapidly from 2010 to 2019, especially in the Shanghai and Tianjin regions. Despite the smaller footprint of $GDP_3$ compared to other sectors, the higher economic output per hectare allows $GDP_3$ to show a potential exposure comparable to $GDP_2$ quantitatively. In 2010, a total of CNY 4373.73 billion (~USD 664 billion) of $GDP_3$ was produced in LECZ, a figure of 42.17% of LECZ's GDP and 24.02% of national $GDP_3$. In 2019, $GDP_3$ within the LECZ achieved a substantial increase, reaching CNY 12,321.77 billion (~USD 1770 billion in 2019), growing to 55.01% of the total GDP within the LECZ. Compared to 2010, $GDP_3$ overtook $GDP_2$ as the main source of economic output in the LECZ in 2019. However, as in all previous sectors, the LECZ's $GDP_3$ share of the national tertiary sector remains declining, changing to 22.7%.

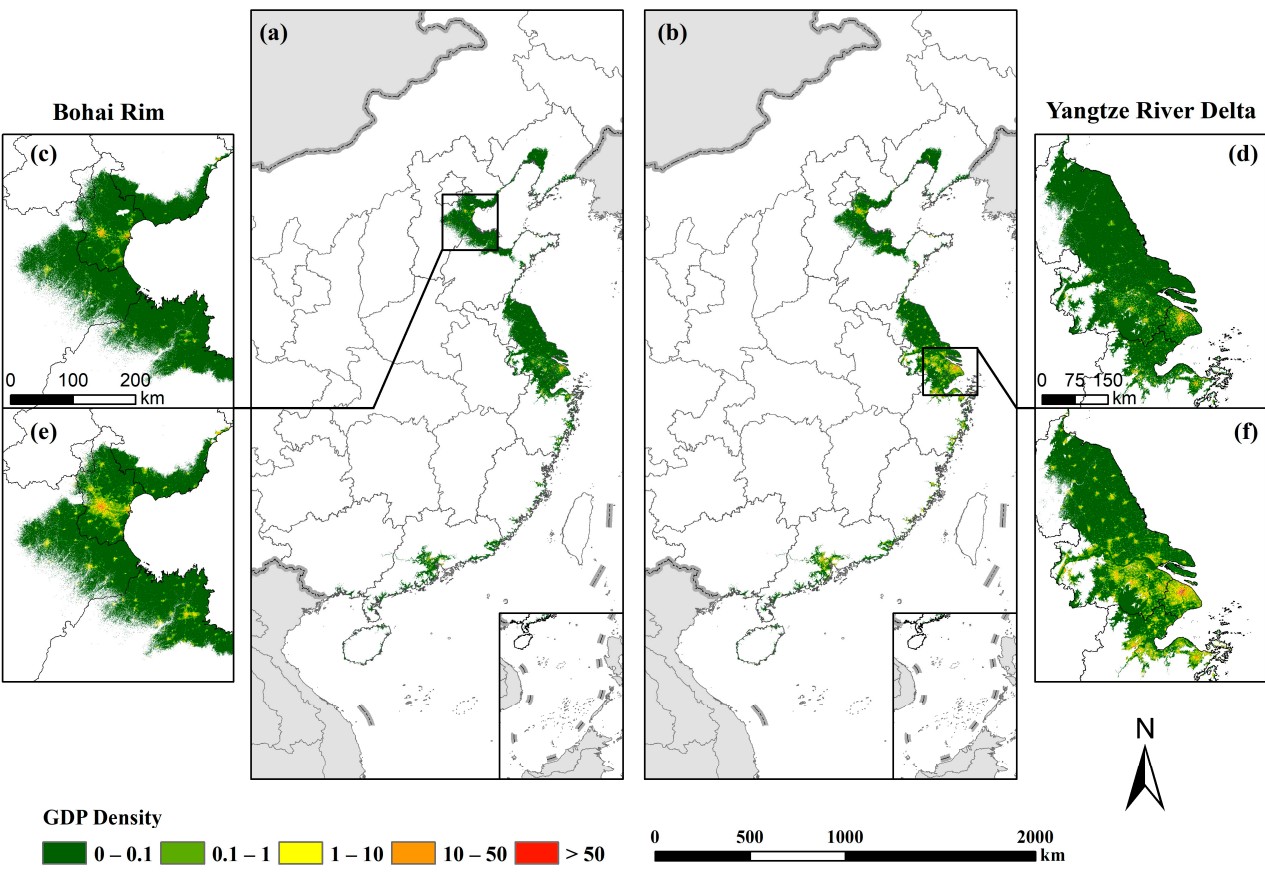

**GDP Density**

| 0 – 0.1 | 0.1 – 1 | 1 – 10 | 10 – 50 | > 50 |

**Figure 7.** A 100 m gridded GDP$_3$ map of China's LECZ in 2010 (**a,c,d**) and 2019 (**b,e,f**) (unit: million CNY/ha).

## 4. Discussion

For a physical event to be hazardous and damaging, a population and/or assets must be exposed to the threat [52]. Previous studies have examined the distribution of population within the LECZ at the global or regional scale and the potential impact of climate change [1,53], but no study has quantified the exposure of GDP in the same manner as population. To better understand the threats facing the LECZ, we first mapped the GDP distribution of three sectors along the coast of China, and then superimposed the LECZ obtained by DEM for GDP assessment. As described above, although the LECZ accounts for only 1.91% of China's territory, the GDP generated within it is enormous (Table 3). Specifically, LECZ contributed 24.02% and 22.7% of China's GDP in 2010 and 2019, respectively. The Yangtze River Delta, the Pearl River Delta and the Bohai Rim are the most economically developed regions in eastern China, but unfortunately are also the regions with the largest LECZ areas. In particular, Jiangsu, Guangdong, Zhejiang and Shanghai are identified as the provinces with the highest GDP output within the LECZ.

All sectors of GDP exhibit significant potential exposure in the LECZ, but behave differently in terms of spatial distribution, aggregate and temporal variation. Nearly 10% of the primary sector and over 20% of the secondary and tertiary sectors are located in the LECZ in mainland China. Not surprisingly, due to the resources and transportation constraints, GDP$_1$ is predominantly located in rural areas, GDP$_2$ is typically generated in suburban fringes, and GDP$_3$ is concentrated in urban centers, especially in highly urbanized metropolises such as Shanghai and Guangdong. However, even with a narrower distribution, higher unit output still determines the economic dominance of GDP$_2$ and GDP$_3$ in the LECZ, whereas GDP$_1$ accounts for less than 5% of LECZ's GDP. In addition, economic output within the LECZ has grown significantly across all sectors from 2010 to 2019, with more economic activity occurring within the LECZ in China (Figure 8). Where

GDP$_1$ growth occurs mainly in the outer urban zones, urban expansion leads to a decline in GDP$_1$ in the former urban fringe. Metropolitan centers and areas far from cities show a slight decrease in GDP$_2$, while suburban areas show a significant increase in GDP$_2$. It is noteworthy that the density of GDP$_2$ in the eastern part of Tianjin shows a significant decrease, which is consistent with the statistical trend of Tianjin. GDP$_3$, which is dominated by service industries, has increased significantly as a share of the economy in the LECZ, replacing GDP$_2$ as the main source of GDP within the LECZ.

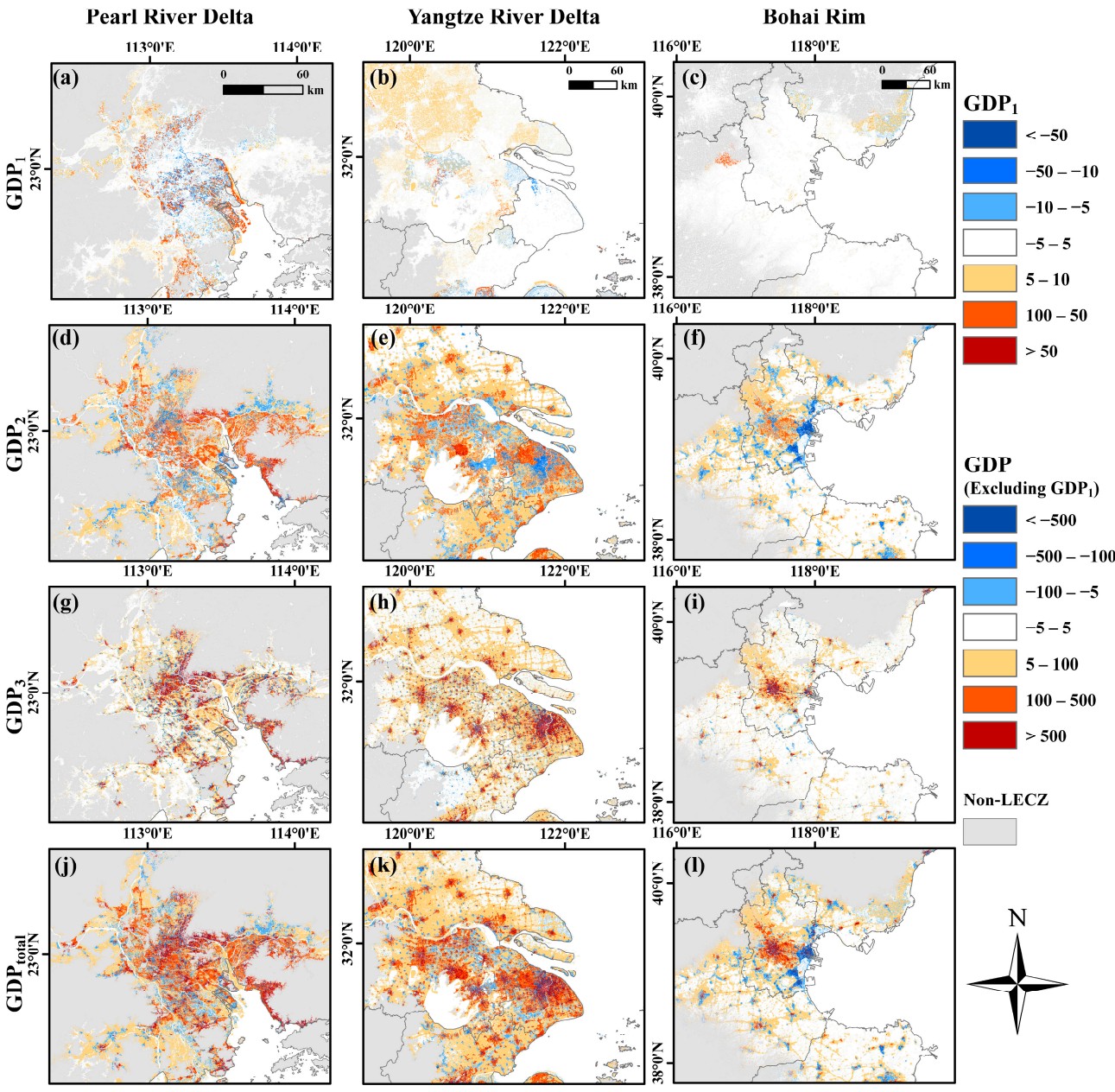

**Figure 8.** Difference between GDP$_1$ (**a–c**), GDP$_2$ (**d–f**), GDP$_3$ (**g–i**) and GDP$_{total}$ (**j–l**) in 2019 and 2010 for the three economic regions of eastern China (million CNY/ha), with positive values indicating GDP growth and negative values indicating decline.

Benefiting from geographical advantages and preferential policies, China's coastal areas, especially the LECZ, have attracted a wealth of population and enterprises in recent decades [54,55]. The clustering of the population and economy in the LECZ further contributes to the rapid development of China. A comparison of our GDP dataset with the LandScan population dataset confirms the high geographic overlap of economic

and demographic hotspots within the LECZ, especially between $GDP_3$ and population (Figure 9). However, a dramatic difference between population clustering and GDP clustering in the LECZ is also observed, with GDP exhibiting stronger regional heterogeneity. McGranahan et al. [1] and Yang et al. [38] have derived that about 10% and 12.7% of the population of mainland China live in the LECZ, respectively. In our results, even $GDP_1$, which has the lowest absolute and relative values within LECZ, yields result similar to the proportion of LECZ's population in the country, whereas $GDP_2$ and $GDP_3$ are almost twice as large as the population (Table 3). Although previous research has shown that wealthier regions tend to have greater capacities to recover from disasters [56,57], for pre-disaster prevention, the higher agglomeration of the economy also implies higher exposure risk, especially within the LECZ in the context of climate change [58].

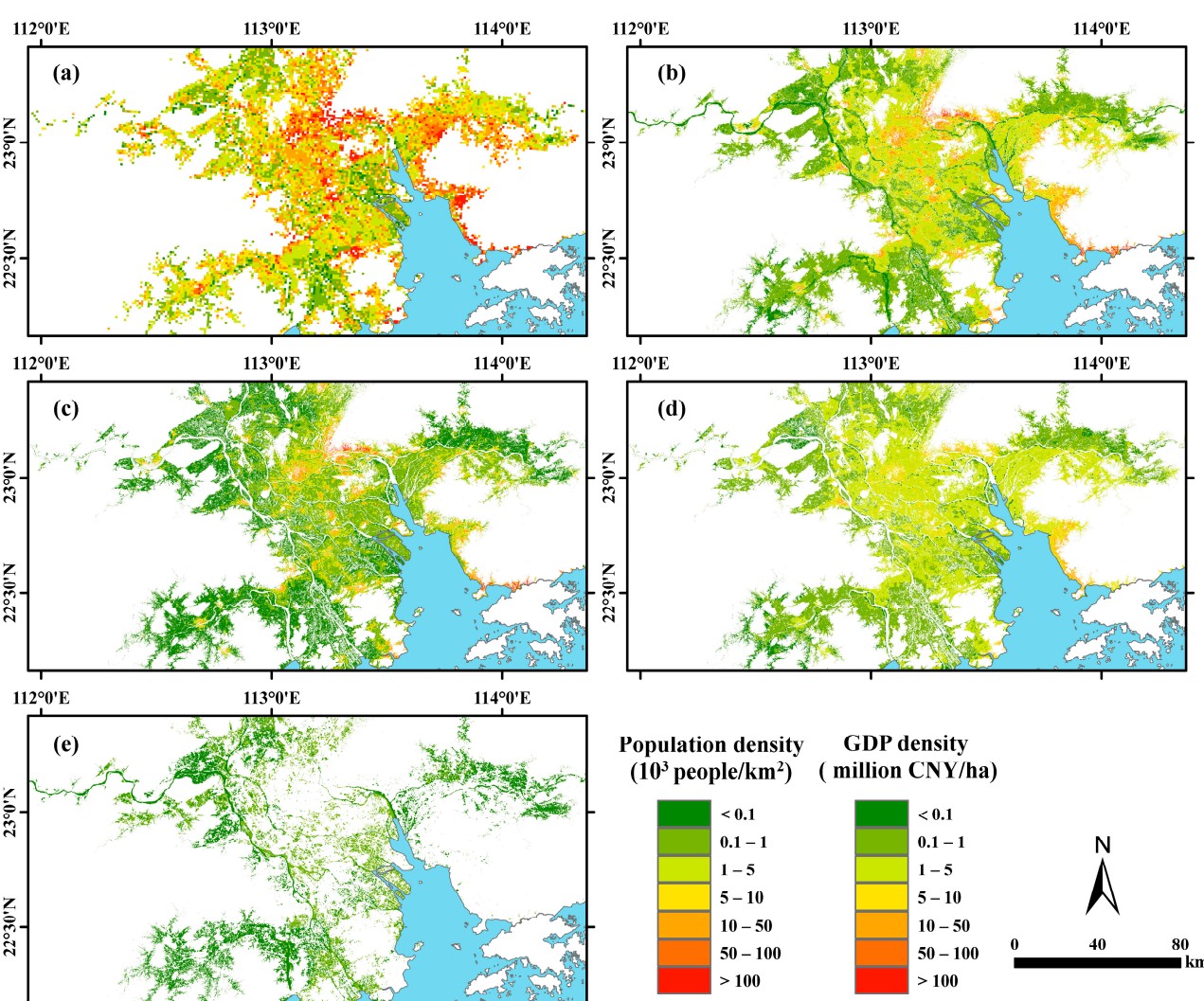

**Figure 9.** Local comparison of LandScan population (**a**) with total GDP (**b**), $GDP_3$ (**c**), $GDP_2$ (**d**) and $GDP_1$ (**e**) within the Pearl River Delta LECZ in 2019. The resolution of LandScan is 1 km and the resolution of GDP data is 100 m.

Moreover, the high degree of population and economic overlap does not mean that it is conducive to concentrated disaster risk prevention within the LECZ. The complexity of the characteristics of economic activities still plagues city managers in responses to disaster threats across sectors [59]. Specific, well-targeted defense policies need to be proposed, as well as the careful consideration of long-term coastal planning options. For example, for existing $GDP_1$ and $GDP_2$, physical defenses such as strengthening the existing protection facilities, building new storm defense facilities and developing coastal shelter

forests will be clear and effective solutions [60,61]. At the same time, special attention should be paid to the fact that the destruction of coastal industries by natural disasters may further trigger serious industrial technological disasters, such as fire, explosions, poisoning, etc., causing a more profound negative impact on society and the environment [20,62]. Moreover, for $GDP_3$ in the LECZ, while the negative effect of climate change and sea level rise are obvious [63,64], measures to address such negative impacts are much more complex. Macroeconomic adjustment to reduce $GDP_3$ exposure in LECZ remains an effective tool for the government department, although this often takes a long period of time [65]. It is important to note, however, that moving all residents and assets out of LECZ is hardly practical so, similar to the population issue, "defense" and "retreat" are the two baseline strategies for addressing the risk to economic activities in the LECZ [53].

Policy preferences are an important factor in China's economic development and will also be an effective strategy to deal with the potential exposure of GDP in the LECZ. This can be explained by another interesting phenomenon observed in this study and contributes to the analysis of the relationship between population and economy in China. As mentioned earlier, firstly GDP (as well as $GDP_1$, $GDP_2$ and $GDP_3$) within the LECZ shows a trend of significant growth in total, but both $GDP_1$ and $GDP_2$ show a decrease in their share of the economy of the LECZ, with $GDP_3$ replacing $GDP_2$ as the main economic source within the LECZ. The transformation of industries within the LECZ in China is ongoing [66]. At the same time, the share of GDP in all sectors of the LECZ is decreasing nationally. Considering the dependence of $GDP_1$ on environmental resources, we believe that the center of gravity of China's $GDP_2$ is gradually migrating to the interior of China, followed by the shift of $GDP_3$ [67,68]. However, LandScan's results show that China's population has maintained its migration to the LECZ during this period. This phenomenon stems from the increasing importance of the inland in relation to the transformation of China's economic reform and open-door policy to the strategy of developing eastern, central and western China simultaneously [69,70]. We make a hypothesis based on this phenomenon that policies drive economic transfer, economic transfer drives population migration and there is often a delay between conversions, primarily between the population and economy. For the consideration of mitigating the impact of climate change, the Chinese government's inward migration policy will help reduce the risk faced by LECZ, but it will be a slow process. The economic status of LECZ will remain difficult to replace for a long time to come.

Although we estimated the potential GDP exposure within the LECZ for China in 2010 and 2019, inevitably multiple limitations remain. Firstly, difficulties in data availability limit our ability to explore more years and regions of potential GDP exposure within the LECZ. More variation between years would be useful to further analyze the spatio-temporal pattern of GDP variation across sectors within the LECZ. This direction has been little studied and more peers are expected. The relationship between population and GDP within the LECZ and national policies also deserves more in-depth exploration. Although we propose a hypothesis from a policy perspective, more detailed data analysis such as geographic detection [71], and spatial autocorrelation methods would help to reveal the underlying causes of the induced changes. Also, even with the current state-of-the art GDP mapping method, our simulation of $GDP_1$ is still under-optimal. The spatialization of $GDP_1$ is still one of the difficulties in the field, and more accurate assessment schemes need to be developed, both in terms of new covariates and fitting methods. In addition, the uncertainties introduced in the assessment process were ignored, and the shoreline changes, DEMs from different sources and the resolution of covariates will effectively affect the estimation results and need to be considered in future more refined and small-scale studies. There is currently a need for a unified and rational assessment scheme for LECZ to support the formulation of disaster mitigation policies.

## 5. Conclusions

In this study, we reveal for the first time the spatial distribution patterns of different economic sectors within China's LECZ by superimposing new gridded GDP maps and

DEM data. Our estimates indicate that the GDP within the LECZ accounted for 24.02% and 22.7% of China's total GDP in 2010 and 2019, respectively, while their corresponding areas accounted for only 1.91% of the country's total area. Similar to the previous studies on population exposure, GDP also showed agglomeration within the LECZ, and even higher aggregation than population. At the same time, the substantial differences in the spatial-temporal distribution, aggregate and economic characteristics of GDP of various sectors make it harder to prevent the risks posed by GDP in the LECZ. Therefore, we are insistent on ensuring that the GDP exposure in the LECZ should be given the same attention as the population, and that sector-specific defense strategies must be developed. Through the assessment of the potential exposure of GDP within China's LECZ, this study highlights the huge risks posed by climate change to economic activities within the LECZ, and the same issue is also worth discussing in other coastal countries.

**Author Contributions:** Conceptualization, X.Y.; methodology, F.L., Q.C. and X.Y.; software, L.M.; validation, L.M. and Q.C.; formal analysis, F.L.; investigation, F.L. and Q.C.; resources, L.M. and Q.C.; data curation, F.L. and X.Y.; writing—original draft preparation, F.L.; writing—review and editing, L.M. and X.Y.; visualization, F.L. and X.Y.; supervision, Q.C.; project administration, X.Y.; funding acquisition, X.Y. All authors have read and agreed to the published version of the manuscript.

**Funding:** This research was funded by the National Natural Science Foundation of China, grant number 41971019 and the Fundamental Research Funds for the Central Universities, grant number 2019QNA4050.

**Data Availability Statement:** All data presented in this study are available upon request from the corresponding author.

**Acknowledgments:** The authors are also thankful to the anonymous referees for their comments and suggestions that improved the quality of this paper.

**Conflicts of Interest:** The authors declare no conflict of interest.

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
