# Peer review of "Refined Estimation of Potential GDP Exposure in Low-Elevation Coastal Zones (LECZ) of China Based on Multi-Source Data and Random Forest"

_remotesensing, doi:10.3390/rs15051285_

Round 1

Reviewer 1 Report

This paper makes an accurate estimation of the GDP data of China's LECZ region using the method of random forest and grid, which has certain practical value. There are several problems:

1. Please specify the specific definitions of the three regions of GDP1, GDP2, and GDP3 at the beginning, and give the specific range diagram.

2. Please explain why the two years of 2010 and 2019 are selected for comparison, and whether it is necessary to add several more years to analyze the differences between the changes in the spatial and temporal pattern and the policies. Whether these two years have very special policy or representative significance.

3. In the data validation part, please explain the fusion method of data from different sources and different formats. If all of them become the average data at the township level, what is the meaning of the grid and why the grid size is set to 100m.

4. When comparing the spatial and temporal patterns of GDP, it is suggested to show their differences in the form of difference chart, highlighting the change areas and reasons.

5. For the reason analysis, the author only used the policy reasons to analyze whether there should be data analysis and specific policy support, and whether there should be specific problem analysis through the geographical detector or some other methods, may be not only because of the policy reasons.

Reviewer 2 Report

The manuscript presents a method for estimating GDP at a high resolution and using it to examine the exposure of economic activities to the expected sea level rise. The manuscript can be improved by considering the following.

1. The R-squared result for GDP1 rural is a bit low. This might be due to the use of mainly proxy data except for the cropland rate. Did the authors try any optimization techniques or removal of redundant data?

2. The authors did not indicate that they faced any challenges due to changes in the coastline. The DEM might be stable over a couple of years but the coastline might not be so stable and this will affect the computation of the LECZ. 

3. In figure 1, the color of the elevation 8-10m according to the legend is white portraying the whole of China as being between 8 and 10 meters. 

4. The author(s) should present a couple of maps (for example, POI density) of the data used for estimating the GDP. 

5. The manuscript needs moderate copy-editing. For example in the statement, "In the extremes, the 21st century could even exceed 2m, threatening to inundate large numbers of people and property worldwide [7,8]", what does 2 m mean?

Reviewer 3 Report

The manuscript titled “Refined Estimation of Potential GDP Exposure in Low-Elevation Coastal Zones (LECZ) of China Based on Multi-source Data and Random Forest” (Manuscript ID: remotesensing-2183158) presents a procedure for estimating the potential exposure of Gross Domestic Product (GDP) in the Low Elevation Coastal Zones (LECZ) across China. My questions about the manuscript are:

1.    Why Random Forest was used in this study as machine learning algorithms? Why not the others (SVM for example)? Please explain or give references that support the usage of random forest.

2.    I think the obtained results should be compared with official statistics and the reasons of the differences should be discussed.

3.    How did you determine the variables of GDP1, GDP2 and GDP3? Please explain.

4.    In Figure 2, for better understanding it would be fine to relate the main steps given in the left with the other steps (right) by using colors or shapes. 

Reviewer 4 Report

Dear authors,

The article is presented with a well mature scientific background and quality results were provided.  I feel the article is fit to publish in the present format.

Reviewer 5 Report

I found this article very interesting. Although the thematic scope of this paper is very wide and requires the use of many different methods, they are well documented and the results are in general clearly presented. I also appreciate the way in which the so far achievements have been documented. The article is well written and structured.

I also realize that the GDP model is difficult to validate, but nevertheless I suggest a minor revision regarding the accuracy of the GDP assessment, especially the total GDP (GDP1). The value of R square is rather small so suggest a rather small fit. This should be clarified in the discussion. Moreover, the adoption of 100 m resolution of estimated GDP requires explanation, also in the discussion. The source data used differ spatial resolution (from 30 m to 1 km), which undoubtedly affects the accuracy of the estimation. Please, clarified this issue.

Round 2

Reviewer 3 Report

The revised version of the manuscript and the given answers to questions are sufficient for me.